

# Complex Networks Reveal Teleconnections between the Global SST and Rainfall in Southwest China

Panjie Qiao[1], Wenqi Liu[1], Yongwen Zhang[1,2], and Zhiqiang Gong[3]

[1]Data Science Research Center, Faculty of Science, Kunming University of Science and Technology, Kunming 650500, China;
[2]Department of Physics, Bar-Ilan University, Ramat Gan 52900, Israel;
[3]Laboratory for Climate Studies, National Climate Research Center CMA, Beijing 100081, China

**Correspondence:** Wenqi Liu (liuwenq2215@sina.com)

**Abstract.** Extreme drought events have frequently occurred in Southwest China (SWC) in this century. The rainfall of SWC could be related to several climate systems such as the East Asia monsoon, Indian monsoon and El Niño-southern Oscillation. Also it tightly depends on the variety of landforms and the complex terrain of SWC. Therefore the mechanism and prediction of rainfall in the area have became a difficult and central issue in climatology. Here we develop a novel multi-variable network

method to delineate the relation between the global sea surface temperature anomalies (SSTA) and the rainfall of SWC. Our results show the important degree patterns in the Western, Eastern Pacific Ocean and Indian Ocean, which significantly influence the rainfall in SWC. Particularly the patterns will change with season and connect to some specific subareas within SWC. The strongest teleconnection is observed for the spring rainfall. The underlying mechanism of our observed teleconnection could be related to the large-scale ocean-atmosphere circulations. Moreover, we can identify the time-lag of the teleconnection links

that can potentially improve the prediction of rainfall in SWC.

## 1 Introduction

Rainfall plays an important role in natural hazards such as droughts and floods. Due to the variety of landforms and the complex terrain in Southwest China (SWC), extreme weather is easier to lead to soil erosion, floods and droughts in SWC than other regions in China (Liu et al., 2009, 2010). In recent years, extreme droughts in SWC have occurred frequently (Sun et al.,

2016, 2017; Wang et al., 2016). The study claimed that droughts in SWC could be frequent in the 21st century compared to the last century (Wang et al., 2014). In recent ten years, extreme rainfall events have caused lots of large-scale landslides and mudslides, resulting in a large number of casualties and property losses in SWC (Gao et al., 2017; Wei et al., 2018).

Thus, the local government is very concerned about the frequent extreme weather events associated with droughts and floods in SWC. Also many researchers have pay considerable efforts to explore the mechanism behind the extreme weather events

in SWC. Some studies have shown that the occurrence of droughts and floods in SWC is closely related to the SSTA (Ha et al., 2019; Wang et al., 2018). For example, the study (Li et al., 2011) showed that the anomalies of the subtropical high in the Western Pacific Ocean inhibit transporting water to the SWC leading to the extreme drought in 2006. One (Feng et al., 2014) studied the drought events in SWC from 1951 to 2010 and found that most of events are related to the anomalies tropical





Pacific and North Atlantic SST in dry season(JFM). A series of recent drought events were studied in SWC after 1990s (Tan
et al., 2017). They found that both the Arctic Oscillation and the South Oscillation strongly influence rainfall of SWC. The
study (Wang et al., 2015) also demonstrated that the rainfall of SWC in autumn is affected by the temperature of the tropical
Northwestern Pacific. The inhomogeneous spatial and temporal distributions of rainfall were found in SWC (Shi et al., 2015).
The precipitation in SWC has been found a downward trend since1960, but the occurrence rate of the flood did not decrease
(Shi et al., 2015; Huang et al., 2014; Zhang et al., 2017). However, most of studies discussed the rainfall of SWC only for a
single season. The relation between the rainfall of SWC and the global SSTA has not yet been fully understood for different
seasons.

Complex network methods have being developed rapidly in the past decade (Newman et al., 2010). It provides a powerful
tool for studying topological and dynamical structures in complex systems (Donges et al., 2009, 2011). Complex network
methods also have been successfully introduced to climate science. The geographic sites or grids are taken as the network
nodes, and linear or nonlinear interactions between the nodes are treated as network edge or links. The strength of the link is
quantified by the cross-correlation or synchronization et al. (Barrat et al., 2004; Zemp et al., 2014). The application of complex
network of climate science has improved the prediction of the climate phenomena (Steinhaeuser et al., 2012, 2011; Barreiro
et al., 2011; Donges et al., 2009; Ludescher et al., 2013; Gong et al., 2008). The El Niño phenomenon can be predicted one
year ahead in advance by using the network method (Ludescher et al., 2014). Furthermore, the networks reveal the strong
local influence of the El Niño phenomenon on some regions (Fan et al., 2017). The spatial characteristics of air pollution in
China are also detected by the network method (Zhang et al., 2018). Recently, the network method provided a great insight into
the function of Rossby waves in creating stable, global-scale dependencies of extreme-rainfall events, and into the potential
predictability of associated natural hazards (Boers et al., 2019). The linkage between the global SST and rainfall in SWC
through the network approach has never been studied. Impacts of the key SST regions on the rainfall of SWC have not been
fully revealed.

In this paper, we use the complex network method to analyze the relation between the rainfall of SWC and the SSTA in
different seasons and try to find some possible early warning signals to improve the predictability of the rainfall in SWC. In
the second part we will introduce the data and how to construct the complex network. The main results are showed in the third
part. Finally we give a short summary and discussion.

## 2  Data and Methodology

### 2.1  Data

The data in this paper is obtained from the European Centre for Medium-Range Weather Forecasts (https://www.ecmwf.int/).
We use the daily averaged rainfall in SWC and global SST from January 1979 to December 2017 with a resolution of $2.5° \times$
$2.5°$. The spatial range of SWC are $20°N - 32.5°N$ and $97.5°E - 110°E$ including $N_r = 36$ grid points totally. This area
mainly spans Sichuan Province, Guizhou Province, Yunnan Province, Guangxi Province and Chongqing City. We totally have
$N_s = 6936$ grid points for the global SST. Due to the seasonality of rainfall, we divide the entire time series of rainfall in SWC




into four seasons corresponding to Spring (March-April-May, MAM), Summer (June-July-August, JJA), Autumn (September-October-November, SON) and Winter (December-January-February, DIF).

## 2.2 Methodology

First, we implement the removal of seasonal cycle to obtain the time series of the SSTA and rainfall anomalies (Fan et al., 2017). We then construct a directionally weighted network. The network nodes $i$ and $j$ can be classified into two subsets by the two different variables respectively; one subset includes the rainfall nodes over SWC and the other includes the global SSTA nodes. $X_i(t)$ is a time series of rainfall anomalies for a node $i$ of SWC, where $t$ spans all years 1979-2017 in a specific season. Here we consider that the SSTA $Y_j(t+\tau)$ has a time lag $\tau$ with $X_i(t)$. The cross-correlation function is written as (Fan et al., 2017):

$$\hat{C}_{ij}(\tau) = \frac{\langle X_i(t) \cdot Y_j(t+\tau) \rangle - \langle X_i(t) \rangle \cdot \langle Y_j(t+\tau) \rangle}{\sqrt{X_i(t)^2} \cdot \sqrt{Y_j(t+\tau)^2}}, \tag{1}$$

where $-\tau_{max} \leq \tau \leq \tau_{max}$ is the time lag, $\tau_{max} = 200$. $\langle \ \rangle$ is averaged for all $t$. We identify the largest absolute value of $\hat{C}_{ij}(\tau)$ and denote the corresponding time lag $\tau^*$. The correlation between sites $i$ and sites $j$ is defined as $C_{ij} = \hat{C}_{ij}(\tau^*)$. If $\tau^* \neq 0$, the link between $i$ and $j$ is directional. The direction of link is from $i$ to $j$ when $\tau^* > 0$ (from $j$ to $i$ when $\tau^* < 0$).

For the definition of the adjacency matrix of the network, a threshold $\Delta$ of correlation is introduced to exclude noise (Zhang et al., 2018). The adjacency matrix is defined with the threshold as

$$A_{ij} = \begin{cases} 1 & |C_{ij}| \geq \Delta \\ 0 & |C_{ij}| < \Delta \,. \end{cases} \tag{2}$$

The importance of nodes in the network is usually quantified by the degree (Fan et al., 2017; Zhang et al., 2018). Since we focus on the influence of the SSTA to rainfall, only the out-degree is considered here. The weighted out-degree of the node $j$ is defined as

$$G_j = \sum_{i,\tau^* \leq 0} A_{ij} C_{ij} \,. \tag{3}$$

$G_j$ can be divided into the positive and negative weighted out-degrees $G_j^p$ and $G_j^n$ by the weight $C_{ij}$ respectively, $G_j = G_j^p + G_j^n$, since positive and negative one can identify the different correlation characteristics.

To observe the important nodes in SWC that are influenced by the SSTA, the weighted in-degree of the node $i$ is defined as

$$H_i = \sum_{j,\tau^* \leq 0} A_{ij} C_{ij} \,. \tag{4}$$

## 3 Result

According to the Eq. 1, we obtain $C_{ij}$ for any pair of nodes between the SSTA and rainfall. Fig. 1 shows the probability distribution function (PDF) of the correlation $C_{ij}$. We find two separated peaks corresponding to positive and negative correlations

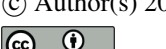
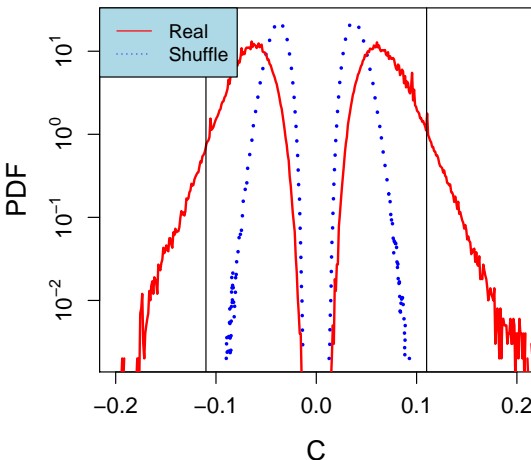

**Figure 1.** (Color online) PDFs of correlations $C_{ij}$ for the real data and shuffle data. Black vertical lines represent the location of the threshold $|\Delta| = 0.11$.

**Table 1.** Statistics of the positive and negative links from the SST to rainfall in SWC for different seasons. $N_p$ and $N_n$ are the total numbers of the positive and negative links respectively. $\langle C_p \rangle$ and $\langle C_n \rangle$ are the averaged correlation measures. $\delta_p$ and $\delta_n$ are the standard deviations of correlations.

|                     | DJF     | MAM     | JJA     | SON     |
|---------------------|---------|---------|---------|---------|
| $N^p$               | 1512    | 3041    | 1869    | 896     |
| $\langle C_p \rangle$ | 0.1232  | 0.1260  | 0.1167  | 0.1195  |
| $\delta_p$          | 0.0112  | 0.0136  | 0.0100  | 0.0089  |
| $N^n$               | 1118    | 1927    | 827     | 768     |
| $\langle C_n \rangle$ | -0.1244 | -0.1271 | -0.1201 | -0.1182 |
| $\delta_n$          | 0.0133  | 0.0148  | 0.0099  | 0.0073  |

respectively in Fig. 1. In order to verify the significance of the correlation, we compare the PDFs between the real data (red) and
shuffled data (blue) in Fig. 1. The shuffled data is obtained by randomizing the original time series. Thus there is no correlation
between the shuffled time series. The PDF of the real data in Fig. 1 shows a much slower decay with the increased absolute
correlation $|C|$ both in the positive and negative parts. For the shuffled data, the PDF almost decays to zero when $|C|$ is larger
than 0.1. Therefore it makes sense to take the threshold $|\Delta| = 0.11$ to filter out the noise and find the significant correlation.

We then use Eq. 2 to get links of the networks. Table 1 summarizes the statistics of the positive and negative links from the
SSTA to rainfall in SWC for the different seasons. We obtain the total numbers of the positive and negative links ($\tau^* \leq 0$) $N_p$





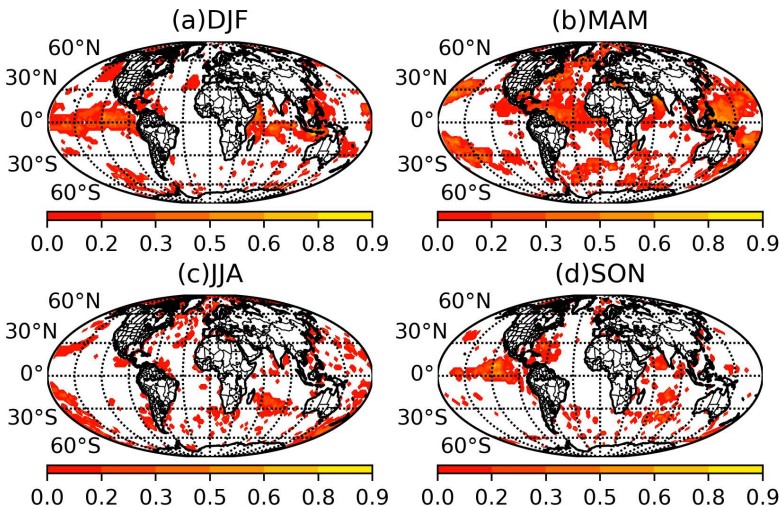

**Figure 2.** (Color online) Distributions of the positive weighted out-degree for different seasons. White areas represents zero in maps.

and $N_n$ respectively in Table 1. Spring rainfall in SWC shows the most positive and negative links connected to the global SSTA. The numbers of the links in winter and summer are less than spring. The fewest links are found in autumn. Similarly the averaged correlation measures $\langle C_p \rangle$ and $\langle C_n \rangle$, standard deviations $\delta_p$ and $\delta_n$ can be calculated for the positive and negative links ($\tau^* \leq 0$), respectively. We also find the most significant values of the averages and standard deviations in spring in Table

1. These results indicate that the influence of the SSTA on rainfall of SWC indeed depends on seasons.

    In order to further obtain the critical regions of the SSTA to affect the rainfall of SWC, we observe the out-degree pattern over the global that is calculated by Eq. 3. The out-degree patterns show some important regions on the oceans to influence the rainfall of SWC for different seasons in Figs. 2 and 3. In winter, the largest cluster for the positive weighted out-degree is located in the middle and East Equatorial Pacific in Fig. 2(a). In spring, the regional size connected to the rainfall in SWC

is greater than other seasons. It includes the tropical Western Pacific and Atlantic regions in Fig. 2(b). We also note that the largest cluster in the middle and East Equatorial Pacific (Fig. 2(a)) disappears in Fig. 2(b). The positive weighted out-degree pattern in winter seems to be opposite to spring. The reversed phase from winter to spring was not reported before. For summer and autumn, Figs. 2(c) and 2(d) depict the clusters with a smaller scale than winter and spring. The distributions of the negative weighted out-degree are shown in Fig. 3. Instead, the largest negative correlation cluster in the middle Atlantic is found for

winter (Fig. 3(a)) and for spring (Fig. 3(b)), the largest negative correlation cluster is located in the Middle and East Equatorial Pacific. It is reasonable that most of the clusters are found in the tropics, since the Hadley circulation plays an important role to transport moisture from the tropics to SWC (Zhao et al., 2009; Gong et al., 2018). We also find that the the Atlantic, West and East Pacific influence SWC significantly in an agreement with the previous studies (Zhao et al., 2009; Zhang et al., 2017; Feng et al., 2014). These three regions are associated with the NAO and ENSO phenomena.




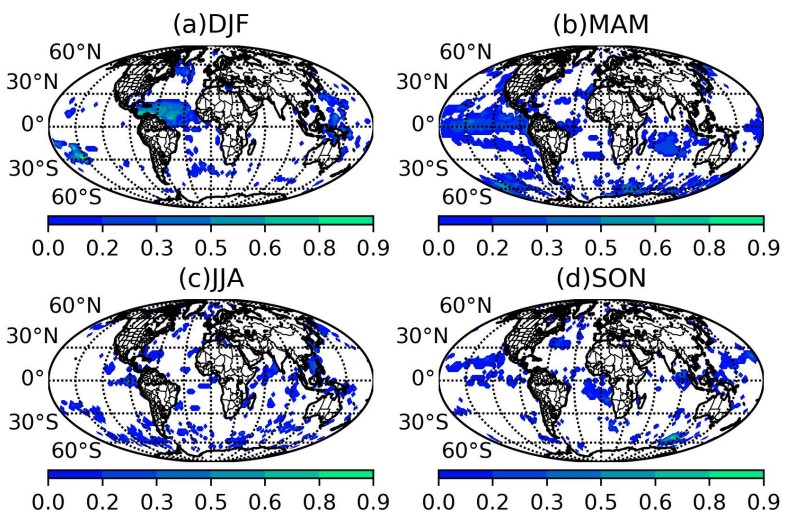

**Figure 3.** Same as FIG. 2 but for negative weighted out-degree.

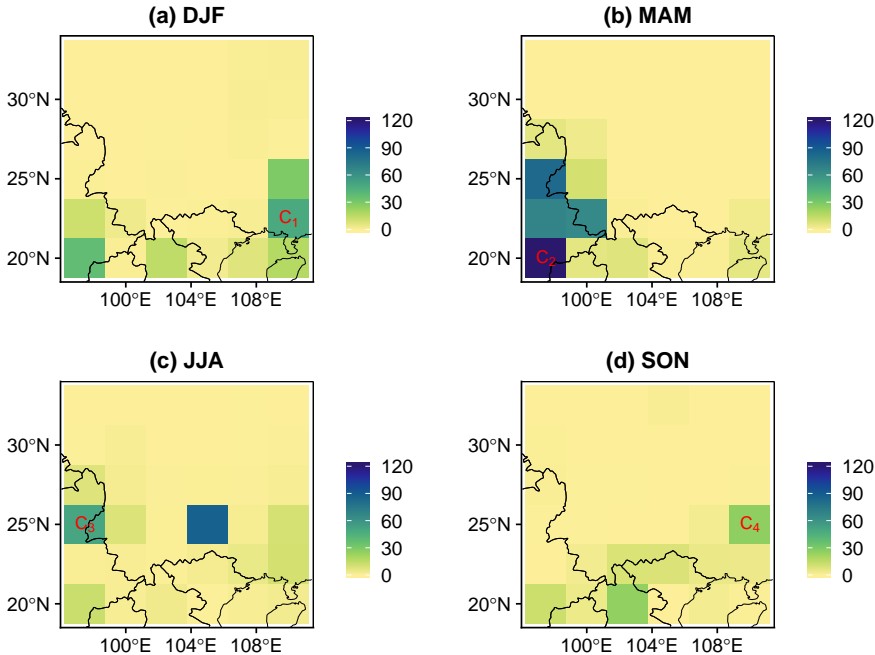

**Figure 4.** (Color online) Distributions of the positive weighted in-degree for different seasons. $C_1$, $C_2$, $C_3$ and $C_4$ are some important nodes of SWC to be influenced by the SSTA.

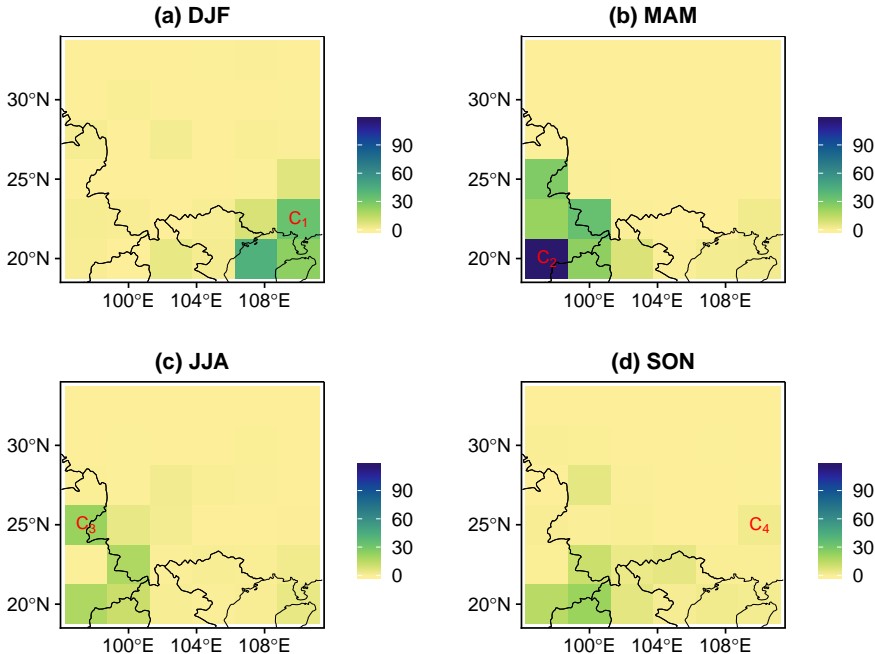

**Figure 5.** Same as FIG.4. but for the negative weighted in-degree.

In the previous studies (Ha et al., 2019; Wang et al., 2016; Li et al., 2011), the whole SWC region was usually treated as a point to be influenced by the SSTA. However, some studies already showed an inhomogeneous spatial distribution of rainfall in SWC (Shi et al., 2015; Ma et al., 2013). To further check the teleconnection between the SSTA and rainfall in SWC associated with the inhomogeneous spatial distribution, we calculated the distribution of weighted in-degree by Eq. 4. Some nodes within SWC show a weak in-degree and only several nodes have strong correlations with the SSTA as shown in Figs. 4

and 5. Importantly, the distributions are localized and change with seasons i.e., the node with the largest in-degree represented as $C_1$ is located at right-bottom corner in Fig. 4(a) for winter; but for spring, the strongest node will change to $C_2$ located at left-bottom corner in Fig. 4(b). Furthermore, the distributions of the positive and negative weighted in-degree are similar (see Figs. 4 and 5). This indicates that if the nodes of SWC are positively correlated with the nodes of the SSTA, on the other hand these nodes in SWC will be negatively correlated with the other nodes of the SSTA.

Next, we show the examples of links between the important nodes of SWC and the SSTA. Here we focus on winter and spring since the stronger correlations are observed in winter and spring as revealed above. Fig. 6(a) shows the links between the rainfall node $C_1$ (see Fig.4(a)) and the SSTA nodes in the East Equatorial Pacific with the positive correlation. The correlation $\hat{C}$ changes with the time lag $\tau$ and reaches to the maximum value at $\tau = -40$ days in Fig. 6(b). It means that a high daily SSTA in the East Equatorial Pacific is probably observed nearly 40 days before a high rainfall rate in SWC. All of the links in

Fig. 6(a) show a similar trend in Fig. 6(b). This could be related to the ENSO phenomenon leading to an abnormal temperature

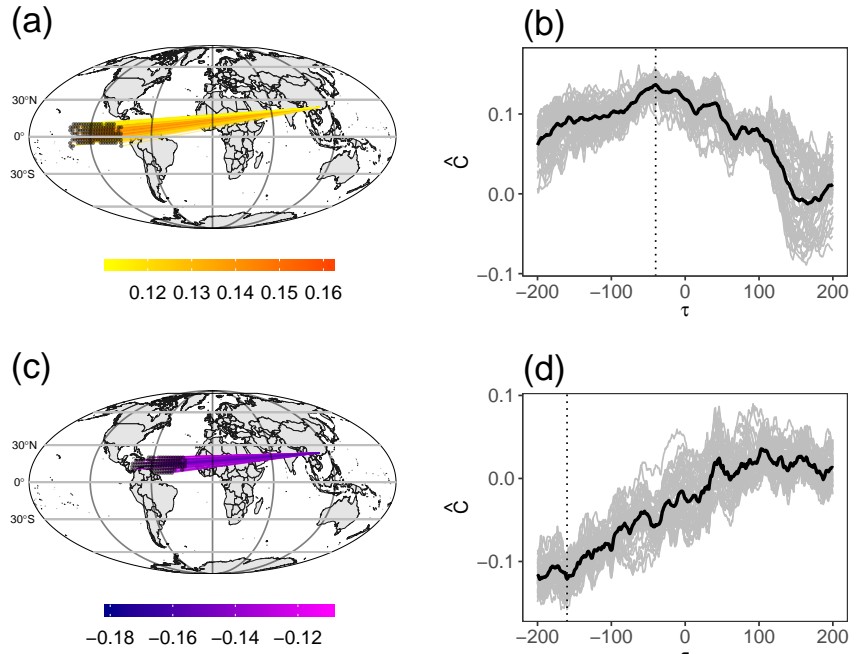

**Figure 6.** (Color online) Examples of the links between the rainfall node $C_1$ and the SSTA nodes (a) in the East Equatorial Pacific with the positive correlation, (c) in the North Atlantic with the negative correlation for DJF (winter). The correlation $\hat{C}$ as a function of the time lag $\tau$ (b) corresponding to the links in map (a) and (d) corresponding to the links in map (c). Grey grey curves represent the different links. Black curve is the average of all grey curves. Dashed black line shows the absolute maximum of the correlation $\hat{C}$ for the average.

in the East Equatorial Pacific, then causing extreme weather events over the global including SWC (Zhao et al., 2009). We next see the negatively correlated links between the nodes $C_1$ and the North Atlantic (see Fig. 6(c)). The absolute maximum of the correlation $\hat{C}$ corresponds to $\tau = -160$ in Fig. 6(d), which has been closed to the limit of the time lag. If we extend the limit, we will clearly observe a trough at $\tau = -160$. These links with the negative correlation (Fig. 6(a)) are earlier to influence the node $C_1$ than the positive links (Fig. 6(c)). The SSTA of the North Atlantic can arouse a series of quasi-stationary wave trains that propagate eastward leading to emerge anticyclone (cyclone) in upper air of SWC so that the rainfall of $C_1$ decreases (increases) (Zhang et al., 2017).

Figs. 7(a) and (c) show the links between the rainfall node $C_2$ (represents in Fig. 4(b)) and the SSTA nodes in the West Equatorial Pacific with the positive correlation and in the East Equatorial Pacific with the negative correlation for spring respectively. The corresponding time lags of the absolute maximum of the correlation $\hat{C}$ are $-95$ and $-74$ days for Figs. 7(b) and (d) respectively. They are close to each other. We conjecture that the correlation between the node $C_2$ and the East Equatorial Pacific could be indirect. The West Equatorial Pacific is intermediary. The relationships between the West Equatorial Pacific and the East Equatorial Pacific are built by the Walker Circulation (Zhang et al., 2017; Yun et al., 2010). The change of convective force with the SSTA in the West Equatorial Pacific can directly affect the moisture transporting from low latitudes





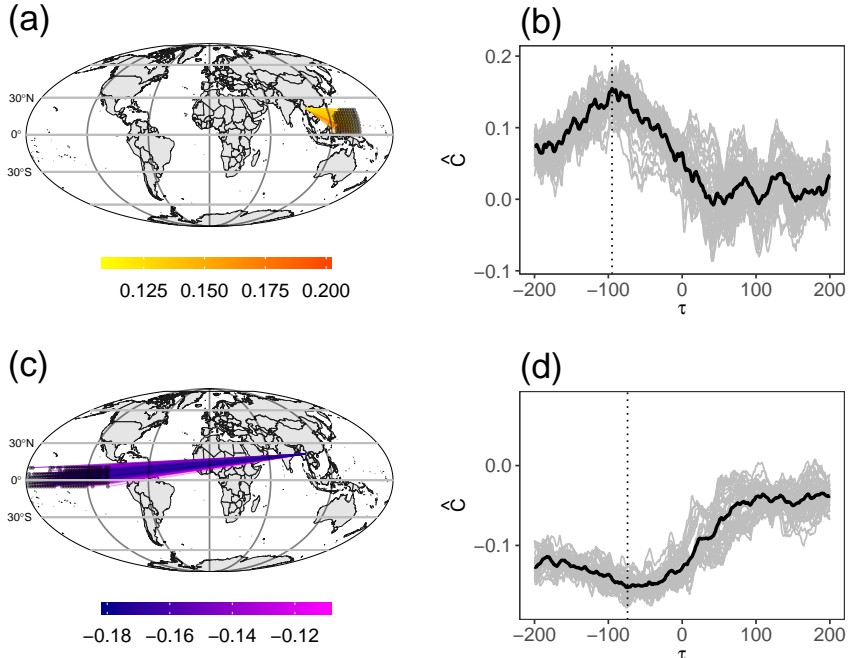

**Figure 7.** (Color online) Same as Fig. 6 for the examples of the links between the rainfall node $C_2$ and the SSTA nodes within (a) the West and (c) East Equatorial Pacific for MAM (spring).

to SWC (Zhang et al., 2017; Feng et al., 2014). However, the impacts of the West Equatorial Pacific on the spring rainfall are localized where the area is around the node $C_2$. Figs. 8 and 9 show the examples of the links for summer and autumn respectively. The Indian Ocean and the North west Pacific are important regions to impact the node $C_3$ for summer. Some studies also suggested that considering both the Indian Ocean and the Western Pacific can improve the prediction of the summer rainfall in east Asia (Cao et al., 2013; Wang et al., 2012).

**4   Conclusions**

In summary, we develop a complex network method to study the teleconnection between the SSTA and rainfall of SWC. Statistics of the networks indicate that the results are strongly depend on seasons. Most teleconnection links are observed in spring both for the negative and positive correlations. The second one is in winter. Summer and autumn have less links. It is reasonable that rainfall is more dependent on transport of moisture from some remote regions in winter and spring, since less

moisture is generated by local sources in SWC. The remote regions (i.e. the tropical Western Pacific, North Atlantic and Middle and East Equatorial Pacific) connected to SWC are observed by the weighted out-degree patterns. Indeed, the teleconnection patterns change with seasons.


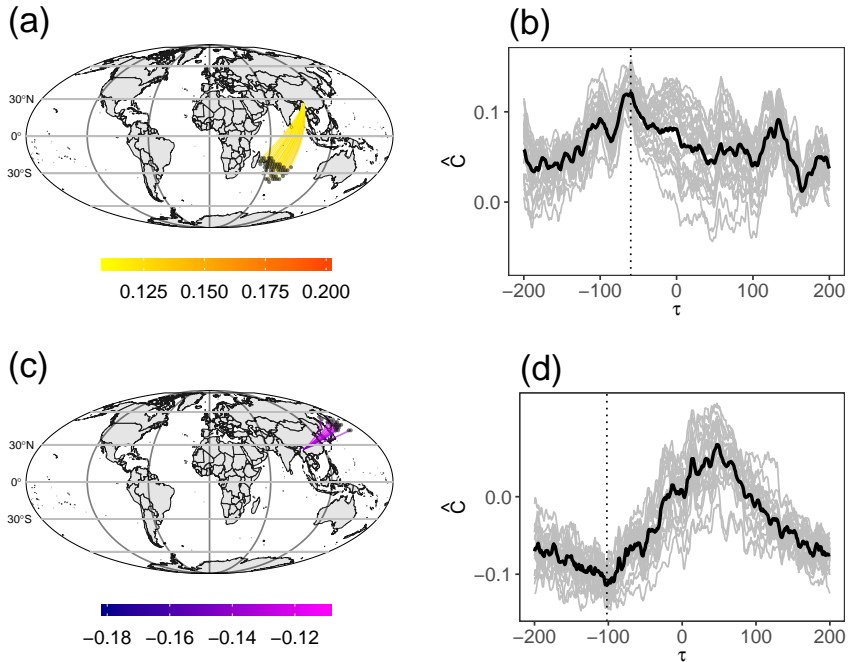

**Figure 8.** (Color online) Same as Fig. 6 for the examples of the links between the rainfall node $C_3$ and the SSTA nodes for JJA (summer).

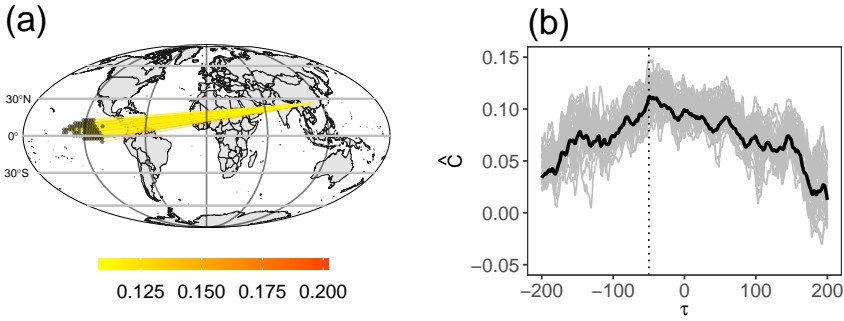

**Figure 9.** (Color online) Same as Fig. 6 for the examples of the links between the rainfall node $C_4$ and the SSTA nodes for SON (autumn).



Due to the inhomogeneous spatial distribution of rainfall associated with the complex topography in SWC, we investigate the distribution of the weighted in-degree in SWC. We find that most of teleconnections are contributed by the several specific

nodes with in SWC and these nodes are different for different seasons. Thus we suggest that it is inappropriate to treat SWC as a homogeneous region. At last, we show some significant links between the SSTA nodes and the rainfall nodes within SWC. We identify the time-lag corresponding to the strongest correlation that can potentially give how many days the nodes within SWC can be influenced by the SSTA i.e., the rain rate in the subareas of SWC could be influenced by the daily SSTA of the North Atlantic 160 days ago in winter and for the West Equatorial Pacific it will be 95 days ahead of the rain of SWC in spring.

We believe that these results will help to improve the prediction of rainfall in SWC.

*Data availability.* The data in this paper was downloaded from the European Centre for Medium-Range Weather Forecasts (https://www.ecmwf.int/).

*Author contributions.* Liu, Gong and Zhang designed research; Qiao performed research; Qiao and Zhang analyzed data; Qiao, Liu, Zhang and Gong wrote the paper.

*Competing interests.* We declare no conflict of interest.

*Acknowledgements.* We are grateful for the financial support by the National Natural Science Foundation of China Project (Grant Nos. 61573173, 41575082 and 41875100). We also acknowledge the data resources provided by the European Centre for Medium-Range Weather Forecasts.



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
