# Peer review of "Complex Networks Reveal Teleconnections between the Global SST and Rainfall in Southwest China"

_Natural Hazards and Earth System Sciences, 2019_

## Referee Comment (RC1) · Anonymous Referee #1 · 18 Oct 2019

General: The general idea of the paper seems relevant and interesting to me. Complex networks are a reasonable choice to study teleconnections in climate systems. Their use in climate science has increased in the last decade, but I am not aware of an application to this region. Furthermore, two-parameter networks are still a rather novel method in this field. However, there have already been several studies linking SST and rainfall in China using different approaches (Zhou et al. 2010, Wu et al. 2012). Furthermore, the chosen region is rather small and its special importance (if there is any) was not made clear to me. I do not see a reason speaking against extending the study area to China as a whole.

[Figure]

The authors mention both floods and droughts as hazards that could be better understood based on this work. However, they do not mention droughts after the introduction. Their inclusion into the abstract is therefore misleading. They are also not showing if the correlations they find actually influence the extreme rainfalls that produce floods or whether the effect is only present for low magnitude precipitation. I am therefore not certain, whether this paper (in its current state) fits the scope of a natural hazard journal.

The used data might not be fit to answer all of the questions the authors pose. They mention the complex topography of the study area, but it is unlikely that a 2.5x2.5° grid is sufficient to fully represent this complexity. The MSWEP precipitation dataset (Beck et al. 2016) with a resolution of up to 0.1° and the same temporal range could be better suited for this task. The data is not always described to the necessary extent. It is unclear whether rainfall or rainfall anomalies are studied.

The methods are not fully described in at least two cases. First, the removal of the seasonal cycle is mentioned, but not explained in details. Second, the splitting of the time series into seasons is not completely clear. Does this lead to one time series for each season? How do these look like: 3 months data – 9 months gap – months data - . . ., or a gapless series of the 3 months.

Pearson Correlation is possibly not fit for the data. When explained, the methods are presented in a way that is understandable to a scientific audience. The potential of the complex networks is not fully exploited. Additional network parameters (e.g. betweenness, clustering) could provide further insights and could support the interpretations that the authors make.

The English language of the manuscript is often poor. There are several (> 50) cases of missing words, typos, grammatical mistakes and poor wording. In some cases, this leads to poor understandability. In contrast, the mathematical formulae are well written and described.

The title of the paper is misleading to some extent, as most of the grid cells that have a substantial degree lie fully or partly outside of China.

The contents of the figures are well chosen. They do however need visual improvement to maximize information gain and understandability. Especially the color maps need improvements. Most figures could be larger in size, as they are hard to interpret in the current form.

The introduction seems too long and repetitive at times. The discussion of the results could be more thorough. Apart from that, the overall length of the paper seems fitting.

Specific:

I have the suspicion that parts of the presented correlation could be caused by common seasonality in the compared parameters. This is supported by the fact that some of the timelag-correlation plots show a minimum and a maximum that are offset by $\sim$180 days (Fig. 6b and 8d). The relationship mentioned in lines 118-119 hints at this as well. Due to this I would appreciate a larger maximum timelag ($\pm$365 days) as well as example plots and statistics for rainfall and SSTA.

Instead of using shuffling for the definition of the threshold, I would suggest a classic 95th percentile significance test combined with a multiple testing correction (e.g. Benjamini-Hochberg). Furthermore, Spearman Rank Correlation is a more fitting measure, as the data is likely non-linear.

Uncertainty bounds should be stated with each of the derived timelags, as these are likely up to $\pm$40 days in some cases (e.g. Fig. 7d).

Technical:

I will not spellcheck the whole manuscript. A very frequent mistake is the lack of "the" in front of words that require it (e.g. lines 6, 23, 25) or its unnecessary presence in other cases (e.g. lines 16, 28). Verb tenses (e.g. lines 4, 19) and prepositions are two other major problems. I advise the authors to make use of professional spell-checking.

The color bars of Fig. 2 and 3 should scale linearly. A higher contrast between the different colors would enhance interpretability. Fig. 4 and 5 could need an overview map, of where in the world this is.

---

## Referee Comment (RC2) · Anonymous Referee #2 · 21 Oct 2019

Overall comments:

1. This paper adopts the complex network method [*Fan et al.*, 2017] to find the seasonal relation between the global sea surface temperature anomalies (SSTA) with the rainfall in southwest China (SWC). But comparing the governing equations used in this study to *Fan et al.* (2017), I feel some modification is done without explanation in the manuscript. In addition, similar (even more advanced) approach and extended application have been presented in quite some studies, e.g. [*Liess et al.*, 2014; *Lu et al.*, 2016]. The authors chose a small region and used a coarse resolution (2.5 X 2.5), which might not work for this approach and the research questions they intended to answer. My understanding is the such complex network needs a vast amount of data to feed, otherwise, the results learnt might be biased. I would suggest the study to extend to a large region to fully utilize that method. Also the authors might want to consider extend their literature reviews on the complex network.

2. The authors chose NHESS to publish their paper; however, I find the scope of the study does not fit well with the journal unless the authors improve their writing to emphasize that. Simply speaking, rainfall in SWC does not necessarily indicate hazards, unless it is extremely – dry or wet.

3. The manuscript needs a substantial improvement on the language. The manuscript is very hard to follow and understand, many sentences are ambiguous. Figures and legends are unclear and misleading. The description of the study approach is incomplete and an in-depth discussion from both statistical and physical perspectives is missing in the manuscript.

Major and specific problems that must be addressed before reconsideration are attached below:

Major problems must be addressed:

1. Introduction: The 2$^{nd}$ and 3$^{rd}$ paragraphs in the introduction are literature review on rainfall in SWC and applications of complex network methods respectively, however, both of them just list several related studies without a logical construction. It is quite difficult for readers to follow up, and it does not help leading to the specific research questions of this study. Furthermore, at the end of this paragraph (i.e., Lines 29-31), authors claim that "most of studies discussed the rainfall of the SWC only for single season", therefore this study would like to explore the relation in different seasons. This is not true. There are many studies done by both Chinese scholars and overseas scholars on different seasons, even if different studies might have focus on one or two seasons. So I recommend the authors to remove this claim. I strongly suggest the authors to revise the literature review, most importantly to include appropriate literature for the scientific part (research gaps etc.) and for the method they mainly adopt (complex network).

2. Line 60: As this study discusses the relation between rainfall and SSTA for four seasons independently, why the authors still need removal of the seasonal cycle of rainfall and SSTA data? Also, in fact, I am not clear how the authors did the removal. Please clarify.

3. Line 75-80: Why should we separate the positive and negative degree? Since the authors only mention that they are different characteristics and display the corresponding regions for positive and negative degrees, but the explanation for the underlying mechanisms of these two degrees is lacking. For instance, in Line 100-110, the authors state that most of the clusters locating in the tropics are reasonable because of the important role of Hayley circulation in moisture transportation. But how does Hayley circulation involve in both positive and negative degrees? The authors need to provide clear explanation otherwise it is difficult for readers to follow. And how do the positive and negative degrees contribute to the rainfall in SWC?

4. Lines 82-88: I suggest the authors to clearly state methodology in details in Section 2.2 instead of in the result section. In addition, the authors should pay attention to address the following questions in their revised manuscript: (1) how to obtain the shuffled data? (2) this sentence, "no correlation between the shuffled time series" (Lines 85-86), is quite confusing. Does it mean "no significant autocorrelation for one time series" or "no significant correlation between two different shuffled time series"? (3) it is also unclear why this threshold (i.e., 0.11) is appropriate. At least some sensitive test should be provided to see the effect of choosing 0.11 or close values. It seems that values around 0.1 are all reasonable guess based on Figure 1. Also, I suggest to reorganize the result section into several subsections in line with revised methodology part, in order to have a clearer structure.

5. Line 88-90: the authors try to verify the significance of the correlation through comparing the PDFs for the real data and shuffled data. However, the difference is not that distinct, with the maximum correlation of the real data only reaching 0.2 compared to 0.1 for the shuffled data. Besides, how is the data being shuffled? Just randomize the whole original time series? Or as done in Fan et al. (2017), the time series is shuffled only in year level and the time ordering within a year is unchanged.

6. Figure 2: The maps are too small for the reader to interpret. The same problem is with Figure 3, Figure 6 - 8 (a, c) and Figure 9 (a). In addition, the caption of Figure 2 is not consistent with color bar.

7. L99 – 100: Why is the regional size in spring greater than other seasons? I suggest some explanation should be provided. In fact there are many places that the authors only present the observation from figures without extended discussion or

explanation. It is very important to provide insights rather than just purely stating patterns that can be seen from the figures.

8. Figure 4 and 5: The legend of color bar is missing. The descriptions of these figures are quite confusing (Lines 110 - 119). For instance, $C_3$ is the grid with the largest value in the Figure 4 (c), then why it is chosen there? As in the manuscript, $C_1$ to $C_4$ are selected based on the largest in-degree value (C1 is consistent with Figure 4, but I am not sure about C4). The same problem is also found in Figure 5 (d) – inconsistent with the manuscript. Authors should clarify their statement carefully.

9. Similar to Comment # 5: L113 – 115: What are the relationships between the identified nodes spatial patterns and the inhomogeneous spatial distribution of rainfall in SWC? The authors could elaborate more about the spatial distribution of rainfall in SWC.

10. L115 – 118: What are the possible mechanisms that induces the changes of the spatial distributions of identified nodes with seasons? E.g., the joint-effects of terrain and important SSTA nodes.

11. L118 – 119: The sentence may be inappropriate, please rewrite it. Since one node of SSTA may positively and negatively correlate with different nodes in SWC.

12. Lines 121-125: A significant test for correlations much be done, as the absolute values of correlation in Figure 6 (b, d) is only around 0.1. With all these very weak correlation values, I cannot be convinced by the statement such as "a high daily SSTA in East Equatorial Pacific is probably observed … in SWC". The same problem is also found in the discussion for different nodes (Lines 135-144). And the color bars of Figure 6, 7, 8 (a, c) and Figure 9 (a) are incomplete.

13. Figure 6 a&c: I think there are better ways to select critical SSTa regions, rather than just simply comparing Figure 2 and Figure 3. I think the authors could utilize more advanced method (e.g. in [*Kawale, 2013; Lu et al., 2016*])

14. L141 – 144: Can the authors provide the links between nodes C2, C3 and the SSTA nodes for both MAM and JJA. Because both C2 and C3 are important nodes in spring and summer based on figures 4 and 5. The authors should explain more if the SSTA nodes affecting C2 are different with the nodes affecting C3 in spring or summer.

15. Conclusion part: This section is only a brief summary of the study. The authors are expected to provide an in-depth discussion from the physical perspective, like potential mechanism, instead of just listing some related results from previous studies. I do not see any contribution from this study from reading the conclusion part.

16. When I read the abstract (Lines 9-10), I got interested in the study because the authors claimed that "the time-lag of the teleconnection links ... prediction of rainfall in SWC". After I read this manuscript, I do not see how this study could achieve this, the authors should provide related analysis or discussion to support how this study can improve the rainfall prediction.

Some minor issues:
1. L128: I suggest removing ", which has been closed to the limit of the time lag" unless the authors can evaluate the significance of it.

**Suggested References:**

Fan, J., Meng, J., Ashkenazy, Y., Havlin, S., & Schellnhuber, H. J. (2017). Network analysis reveals strongly localized impacts of El Niño. Proceedings of the National Academy of Sciences of the United States of America, 114(29), 7543–7548. https://doi.org/10.1073/pnas.1701214114

Kawale, J. (2013). Mining Relationships in Spatio-temporal Datasets. PhD dissertation. University of Minnesota Twin Cities, Minneapolis, Minnesota, USA.

Liess, S., A. Kumar, P. K. Snyder, J. Kawale, K. Steinhaeuser, F. H. M. Semazzi, A. R. Ganguly, N. F. Samatova, and V. Kumar (2014), Different Modes of Variability over the Tasman Sea: Implications for Regional Climate, *J. Clim.*, *27*(22), 8466–8486, doi:10.1175/JCLI-D-13-00713.1.

Lu, M., U. Lall, J. Kawale, S. Liess, and V. Kumar (2016), Exploring the Predictability of 30-Day Extreme Precipitation Occurrence Using a Global SST–SLP Correlation Network, *J. Clim.*, *29*(3), 1013–1029, doi:10.1175/JCLI-D-14-00452.1.

---

## Author Comment (AC1) · 31 Dec 2019

We thank the referee for constructive comments and comprehensive analyses of our manuscript. We have fully addressed the comments. For your convenience, we now provide our point-by-point responses to all the concerns as detailed below. Note that the referee's comments are in italic font (blue), whereas our reply is not italicized and some corrections (red) in the revised manuscript are attached. All figures are placed at the end of text.

*General: The general idea of the paper seems relevant and interesting to me. Complex networks are a reasonable choice to study teleconnections in climate systems. Their use in climate science has increased in the last decade, but I am not aware of an application to this region. Furthermore, two-parameter networks are still a rather novel method in this field. However, there have already been several studies linking SST and rainfall in China using different approaches (Zhou et al. 2010, Wu et al. 2012). Furthermore, the chosen region is rather small and its special importance (if there is any) was not made clear to me. I do not see a reason speaking against extending the study area to China as a whole.*

Response: We appreciate that our paper is interesting to the referee. Rainfall in this region is the first time to be studied by using the network approach. Although there have been many studies of the linkages between the SST and rainfall in China, we believe that their approaches are different with ours. Complex networks allow us to detect more physical information. Based on the referee's comments, we renamed the chosen area as Northwestern South Asia (NWSA), since the chosen area is little larger than Southwest China. We chose the area as our study region, because in the summer of 2006 and 2011, NWSA suffered from record-breaking drought events leading to an economic loss of 3 billion dollars, 16 million people cannot easily access to drinking water, more than 50 million people were affected by this disaster and nearly a million hectares could not producing crops(Shi et al., 2015). Droughts and floods have close relation with the rainfall in NWSA, so the investigation of rainfall in this region has been an important topic deserving high attention from the meteorologists. The mechanism of precipitation in NWSA is quite complicated, since the East Asian monsoon, Indian monsoon both potentially influence the rainfall in this region. We improved our text in the revised manuscript.

Corrections:

(Line 1-3) Abstract. Droughts and floods have frequently occurred in Northwestern South Asia (NWSA) in this century. The mechanism of precipitation in NWSA is quite complicated, since the East Asian monsoon, Indian monsoon and et al. potentially influence the rainfall in this region. Prediction of precipitation in NWSA has become a difficult and critical topic in climatology study.

(Line 12-19) In recent decades, natural hazards (such as droughts and floods) have occurred frequently in Northwestern South Asia (NWSA) due to climate change, causing a large number of casualties and property losses (Ha et al., 2019; Gao et al., 2017; Wei et al., 2018). In the summer of 2006 and 2011, NWSA suffered from record-breaking droughts events (Zhang et al., 2017). On the other hand, the portion of annual precipitation contributed by extremely heavy precipitation has been found an increasing trend from 1961–2010 in NWSA (Ma et al., 2013). Due to an increasing population and the high risk of natural hazards, NWSA has attracted lots of attention in meteorological research fields. According to CMIP5 multi-model projections, they found that severe and extreme droughts in NWSA increase dramatically in the future, and extremely wet events will also increase (Wang et al., 2014).

*C1*

*The authors mention both floods and droughts as hazards that could be better understood based on this work. However, they do not mention droughts after the introduction. Their inclusion into the abstract is therefore misleading. They are also not showing if the correlations they find actually influence the extreme rainfalls that produce floods or whether the effect is only present for low magnitude precipitation. I am therefore not certain, whether this paper (in its current state) fits the scope of a natural hazard journal.*

Response: We thank the referee very much for this valuable comment. Extremely high precipitation is closely associated with a flood event in NWSA. Besides, continuous normal may also cause large value precipitation, which has an important impact on the possible flood events. Meanwhile, the drought event in NWSA always caused by the little precipitation in a long duration. Therefore, if we can properly predict the rainfall in a long ahead of time, which will be of great help for us to deal with the relevant drought and flood events. This study mainly focusses on the possible correlations between rainfall in NWSA and global key regions' sea surface temperature (SST) anomalies through the complex network approach, which may also help us to better predict floods and droughts in NWSA. In order to find out the influence of extreme rainfall on the correlation between rainfall and SST, we replaced top and bottom 5% extreme precipitation in the middle random magnitude precipitation in each grid data series. Then we analyzed the data by using our method and compared the results with those before replacing. Fig. 2 and Fig. 3 show that most of the correlation patterns disappear in Fig. 2 (b), (d), (f) and (h) after replacing the extreme rainfall event, indicating that top and bottom 5% extreme precipitation plays important roles to produce the correlation patterns. Therefore, this study revealed the correlation between the PA and SSTA is quite important for the extreme rainfall analyses, which further show the tightly connection with the drought and flood in NWSA. Therefore, content of this study quite fits the scope of a natural hazard journal.

Corrections:

(Line 126-130) To further prove that extreme rainfall is significantly influenced by these important regions, top and bottom 5% extreme PA is replaced by the random middle magnitude PA in data. Then we employ the same analysis of the new time series. Fig. 2(b), (d), (f) and (h) shows the results after replacing. Comparing with Fig. 2(a) before replacing, some important nodes disappear in Fig. 2(b) after replacing. Similar results also can be found for other seasons. It implies that top and bottom 5% extreme precipitation plays important roles to contribute to the teleconnection patterns.

*The used data might not be fit to answer all of the questions the authors pose. They mention the complex topography of the study area, but it is unlikely that a 2.5×2.5° grid is sufficient to fully represent this complexity. The MSWEP precipitation dataset (Beck et al. 2016) with a resolution of up to 0.1° and the same temporal range could be better suited for this task. The data is not always described to the necessary extent. It is unclear whether rainfall or rainfall anomalies are studied.*

Response: We thank the referee for the comment. Although the horizontal resolution of the NWSA rainfall data is $2.5° \times 2.5°$, we focused on studying teleconnection between NWSA precipitation anomalies and the global SSTA, reveal the key SST region influence the rainfall in NWSA, the resolution of for rainfall seems not quite crucial, because the SSTA often has impact on large circulation system, for example the SST's impact often covers the entire NWSA region. We think that the main results with high resolution rainfall data will be same as the present study. In our study, precipitation was detrended to get the precipitation anomalies (PA). The analyses in this study are calculated based on the PA and SSTA, relevant descriptions were introduced in the section 2.2. We claimed it in the revised manuscript.

Corrections:

(Line 60-64) First, we remove the seasonal cycle to obtain the time series of the SSTA as (Fan et al., 2017; Meng et al., 2017),

$$Y^y(t) = \frac{\tilde{Y}^y(t) - \text{mean}\left(\tilde{Y}(t)\right)}{\text{std}\left(\tilde{Y}(t)\right)}, \qquad (1)$$

where $\tilde{Y}^y(t)$ is the time series of the daily SST; y stands year and t stands date within a year. "mean" and "std" denote the mean and standard deviation of the SST for all the years on a date t. We use the same way to obtain precipitation anomalies (PA).

*The methods are not fully described in at least two cases. First, the removal of the seasonal cycle is mentioned, but not explained in details. Second, the splitting of the time series into seasons is not completely clear. Does this lead to on time series for each season? How do these look like: 3 months data – 9 months gap – months data - . . ., or a gapless series of the 3 months.*

Response: Thanks. We explained the details of the removal of the seasonal cycle in the above response. We also improved the explanation about the splitting of the time series into seasons as following corrections.

Corrections:

(Line 68-72) We take 3 months for a season in each year, for example June, July and August are selected for summer. Thus for each grid i in NWSA, we can obtain 117 months daily data for 39 years as the PA time series for a season, $X_i(t)$, where t spans those selected days with 9 months gap for each year. Then the corresponding time series of SSTA can be obtained for as $Y_j(t + \tau)$, where $\tau$ is a time delay. Note that the corresponding time series of SSTA depends on the time delay and could be not in the same season as the time series of the PA.

*Pearson Correlation is possibly not fit for the data. When explained, the methods are presented in a way that is understandable to a scientific audience. The potential of the complex networks is not fully exploited. Additional network parameters (e.g. betweenness, clustering) could provide further insights and could support the interpretations that the authors make.*

Response: We thank the referee for this kind comments. We are considered the additional parameters for our study. We detected and found some significant clusters in our networks. This result was discussed in Section 3.3 of the revised manuscript. We selected the nodes in NWSA with the largest weighted in-degree for each season as showed in Figs. 4 and 5. The largest cluster $C_1$ on the sea for a node of NWSA is defined by the largest successive area where all the inside SSTA nodes are connected to that node in NWSA. We can obtain the second largest cluster $C_2$ in a similar way. The large cluster regions can be of help to choose the key region having a crucial impact on rainfall in NWSA.

Corrections:

(Line 149-153) We first select the nodes in NWSA with the largest weighted in-degree for each season as showed in Figs. 4 and 5. The largest cluster $C_1$ is identified by the largest successive area where all the inside SSTA nodes are connected to that important node in NWSA (Kawale, 2013; Lu et al., 2016). We can obtain the second largest cluster $C_2$ in a similar way. Figs. 6(a) shows the cluster $C_1$ (blue) and $C_2$ (green) which are connected to the nodes of $I_{11}$ (as shown in Fig. 4(a)) for DJF.

*The English language of the manuscript is often poor. There are several (> 50) cases of missing words, typos, grammatical mistakes and poor wording. In some cases, this leads to poor understandability. In contrast, the mathematical formulae are well written and described.*

Response: We thank the referee for the careful comment. We did our best to improve the text of the revised manuscript.

*C2*

*The title of the paper is misleading to some extent, as most of the grid cells that have a substantial degree lie fully or partly outside of China.*

Response: Thanks. In order to avoid misleading, we revised Southwest China as Northwestern South Asia (NWSA) in this study. For clarity, we claimed this in the revised manuscript.

Corrections:

(line 54-55) This area mainly includes Southwest China and its surrounding areas.

*The contents of the figures are well chosen. They do however need visual improvement to maximize information gain and understandability. Especially the color maps need improvements. Most figures could be larger in size, as they are hard to interpret in the current form.*

Response: We thank the referee for this comment. We have improved these figures in the revised manuscript.

*The introduction seems too long and repetitive at times. The discussion of the results could be more thorough. Apart from that, the overall length of the paper seems fitting.*

Response: Thank the referee very much. We have rewrote the introduction and discussion in the revised manuscript.

*Specific:*

*I have the suspicion that parts of the presented correlation could be caused by common seasonality in the compared parameters. This is supported by the fact that some of the time lag-correlation plots show a minimum and a maximum that are offset by -180 days (Fig. 6b and 8d). The relationship mentioned in lines 118-119 hints at this as well. Due to this I would appreciate a larger maximum time lag (±365 days) as well as example plots and statistics for rainfall and SSTA.*

Response: We thank the referee for the helpful comment. To verify the referee's suspicion, we apply a new shuffling testing. We randomly shuffled the order of years, keeping the variations within each year and then calculate the cross-correlation between the shuffled time series. In Fig. 1, we showed the PDF of correlations for the shuffle data comparing with real data. In this shuffling process, the distribution of values and the autocorrelations and common seasonality in each year has been kept in each shuffled record, while the physical dependencies between nodes tend to be destroyed. If the correlations are significantly higher than the significant threshold, we regard it as a true link; otherwise, it is suspected to be a spurious link. We obtained the threshold $\Delta = 0.1$ by using the 95% confidence significance test combined with a multiple testing correction (Benjamini-Hochberg) as you suggested. Thus the presented correlation cannot be caused by common seasonality in the revised

manuscript. We also extended the range of time-lag (±365 days) and obtained the sharp minimums and maximums in Fig. 6-9 of the revised manuscript.

Corrections:

(Line 99-103) In order to verify the significance of the correlation, we compare the PDFs between the real data (red) and shuffled data (blue) in Fig. 1. We randomly shuffle the order of years for each node, keeping the variations within each year to get shuffle data (Fan et al., 2017). Then we calculate the cross-correlation for shuffled data as same as real data. In this shuffling process, the autocorrelations and common seasonality in each year have been kept in each shuffled time series, while the physical dependencies between the SSTA and PA nodes are destroyed.

*Instead of using shuffling for the definition of the threshold, I would suggest a classic 95th percentile significance test combined with a multiple testing correction (e.g. Benjamini-Hochberg). Furthermore, Spearman Rank Correlation is a more fitting measure, as the data is likely non-linear.*

Response: We thank the referee's helpful comment. We implemented new shuffling testing and 95th percentile significance test combined with a multiple testing as above response. For Spearman rank correlation, we need to order the time series of SSTA for each time-lag to calculate correlation which will spend more time to calculate. Based on the reviewer's good suggestion, we will do this in further studies.

*Uncertainty bounds should be stated with each of the derived time lags, as these are likely up to ±40 days in some cases (e.g. Fig. 7d).*

Response: Thank the referee. We extended the range of time-lag and uncertainties bounds in Fig. 6-9 have been stated.

*Technical:*

*I will not spellcheck the whole manuscript. A very frequent mistake is the lack of "the" in front of words that require it (e.g. lines 6, 23, 25) or its unnecessary presence in other cases (e.g. lines 16, 28). Verb tenses (e.g. lines 4, 19) and prepositions are two other major problems. I advise the authors to make use of professional spell-checking.*

Response: We thank the referee for the comment. We modified these errors and did our best to spell-checking in the revised manuscript.

*C3*

*The color bars of Fig. 2 and 3 should scale linearly. A higher contrast between the different colors would enhance interpretability. Fig. 4 and 5 could need an overview map, of where in the world this is.*

Response: We thank the referee very much. We changed the color bars of Fig. 2 and 3 to the scale linearly and higher contrast. We marked the location of NWSA as a purple rectangle in the world in Fig. 2 and Fig. 3. Also we mentioned it in the caption of Fig. 4.

[Figure]

**Figure 1.** (Color online) PDFs of correlations $C_{ij}$ for real data and shuffle data. Black vertical lines represent the location of the threshold $|\Delta| = 0.1$.

[Figure]

**Figure 2.** (Color online) Distributions of the positive weighted out-degree for (a) DJF, (c) MAM, (e) JJA and (h) SON. (b), (d), (f) and (h) Same as (a), (c), (e) and (h) but for replacing top and bottom 5% extreme precipitation with middle magnitude precipitation in data. White areas represent zero in maps. Purple rectangle area covers the region of NWSA.

[Figure]

**Figure 3.** Same as Fig. 2 but for the negative weighted out-degree.

[Figure]

**Figure 4.** (Color online) Distributions of the positive weighted in-degrees for (a) DJF, (b) MAM, (c) JJA and (d) SON in NWSA. The location of NWSA in the world is shown as the purple rectangle area in Fig. 2. $I_{11}$, $I_{31}$ and $I_{36}$ are the important nodes in NWSA with the largest positive weighted in-degree in a season.

[Figure]

**Figure 5.** (Color online) Same as FIG. 4 but for the negative weighted out-degree. $I_{15}$, $I_{11}$, $I_{22}$ and $I_{12}$ are the important nodes in NWSA with the largest negative weighted in-degree in a season.

[Figure]

**Figure 6.** (Color online) Locations of the largest cluster $C_1$ (blue) and the second largest cluster $C_2$ (green) that are (a) positively ((c) negatively) correlated with the node $.I_{11}$, $(I_{15})$ in NWSA for DJF. The blue and green arrows represent the strongest links from $C_1$ and $C_2$ to that node in NWSA respectively. (b), (d) The correlation $\hat{C}$ as a function of the time lag $\tau$ corresponding to the strongest links in map (left) respectively. Dashed black line shows the absolute maximum of the correlation $\hat{C}$.

[Figure]

**Figure 7.** (Color online) Locations of the largest cluster $C_1$ (blue) and the second largest cluster $C_2$ (green) that are (a) positively ((c) negatively) correlated with the node $I_{11}$ $(I_{11})$ in NWSA for MAM. Everything else is the same as Fig. 6.

[Figure]

**Figure 8.** (Color online) Locations of the largest cluster $C_1$ (blue) and the second largest cluster $C_2$ (green) that are (a) positively ((c) negatively) correlated with the node $I_{31}$ ($I_{22}$) in NWSA for JJA. Everything else is the same as Fig. 6.

[Figure]

**Figure 9.** (Color online) Locations of the largest cluster $C_1$ (blue) and the second largest cluster $C_2$ (green) that are (a) positively ((c) negatively) correlated with the node $I_{36}$ ($I_{12}$) in NWSA for SON. Everything else is the same as Fig. 6.

---

## Author Comment (AC2) · 31 Dec 2019

We thank the referee for constructive comments and comprehensive analyses of our manuscript. We have fully addressed the comments. For your convenience, we now provide our point-by-point responses to all the concerns as detailed below. Note that the referee's comments are in italic font (blue), whereas our reply is not italicized and some corrections (red) of the revised manuscript are attached. All figures are placed at the end of text.

*Overall comments:*

1. *This paper adopts the complex network method [Fan et al., 2017] to find the seasonal relation between the global sea surface temperature anomalies (SSTA) with the rainfall in southwest China (SWC). But comparing the governing equations used in this study to Fan et al. (2017), I feel some modification is done without explanation in the manuscript. In addition, similar (even more advanced) approach and extended application have been presented in quite some studies, e.g. [Liess et al., 2014; Lu et al., 2016]. The authors chose a small region and used a coarse resolution (2.5 X 2.5), which might not work for this approach and the research questions they intended to answer. My understanding is the such complex network needs a vast amount of data to feed, otherwise, the results learnt might be biased. I would suggest the study to extend to a large region to fully utilize that method. Also the authors might want to consider extend their literature reviews on the complex network.*

Response: We thank the referee very much for these useful and important comments. Based on referee #1's comments, we renamed the chosen area as Northwestern South Asia (NWSA) in the revised manuscript, since the chosen area is little larger than Southwest China. Indeed there are some differences between Fan's method and ours. Fan et al., 2017 mainly studied the networks for one variable. Here we constructed networks between two variables: the global SSTA and precipitation anomalies (PA) in NWSA. Moreover, networks for four seasons were considered in our paper. We claimed these in the revised manuscript. We read the useful refs (Liess et al., 2014; Lu et al., 2016) and cited them in the revised manuscript. Their methods are worth to learn. In our paper, the time series for each grid is enough length (39 years) to guarantee unbiased. Moreover, for the SSTA, we considered a large spatial size (global). Actually, Fan et al., 2017 also considered the networks between the El Niño region (small size) and global temperature anomalies (large size). Thus we believe that the network method is also fitting for our study. We carefully extended our literature reviews on the complex network in the revised manuscript.

Corrections:

(Line 47-48) In this paper, we use a multi-variable network method to analyze the relation between the global SSTA and precipitation anomalies (PA) in NWSA for different seasons.

(Line 66-72) We then construct the directionally multi-variable network. Network nodes i and j can be classified into two subsets by two different variables respectively; one subset includes the PA nodes over NWSA and the other includes the global SSTA nodes. We take 3 months for a season in each year, for example June, July and August were selected for summer. Thus for each grid i in SWC, we can obtain 117 months' daily data for 39 years as the PA time series for a season, $X_i(t)$, where t spans those selected days with 9 months gap for each year. Then the corresponding time series of SSTA can be obtained for as $Y_j(t + \tau)$, where $\tau$ is a time delay. Note that the corresponding time series of SSTA depends on the time delay and could be not in the same season as the time series of the PA. The cross-correlation function is written as (Fan et al., 2017; Zhang et al., 2019)

(Line 33-46) Some new approaches have been developed to better detect and understand teleconnections in climate (Liess et al., 2014). Such as a complex networks approach has been applied to a wide variety of disciplines in the past decade (Barabasi et al., 1999; Newman et al., 2010; Barrat et al., 2004), which also emerged as a powerful tool to study climate systems (Donges et al., 2009a, b). The geographic sites or grids are taken as network nodes, and linear or nonlinear interactions between the nodes are treated as network edges or links. The strength of the link used to be quantified by a cross-correlation or synchronization et al. (Donges et al., 2009a; Boers et al., 2013). The networks were used to study the teleconnection patterns of the El Niño and North Atlantic Oscillation (Tsonis et al., 2008; Yamasaki et al., 2008; Guez et al., 2013). Applications of complex networks in climate science have improved the understanding of many climate phenomena (Steinhaeuser et al., 2011, 2012; Barreiro et al., 2011; Gong et al., 2008; Fan et al., 2017). The El Niño phenomenon can be predicted one year ahead in advance by using the network method (Ludescher et al., 2013,2014). Some features of air pollution have also been detected by the networks (Zhang et al., 2018, 2019). Recently, the network method provided a great insight into the function of the Rossby waves in creating stable, global-scale dependencies of extreme-rainfall events, and into the potential predictability of associated natural hazards (Boers et al., 2019). Furthermore, extreme precipitation events can be predicted even in long-term scale in some cases by using the networks (Lu et al., 2016; Boers et al., 2014).

2. *The authors chose NHESS to publish their paper; however, I find the scope of the study does not fit well with the journal unless the authors improve their writing to emphasize that. Simply speaking, rainfall in SWC does not necessarily indicate hazards, unless it is extremely – dry or wet.*

Response: We thank the referee very much for this valuable comment. Extremely low and high precipitation is associated with droughts and floods respectively. The correlations between precipitation anomalies (PA) and SSTA may be mainly contributed by extremes. To answer this question, we replaced top and bottom 5% extreme precipitation with the random middle magnitude precipitation in data. Then we analyzed the data by using our method and compared the results with before replacing. Fig. 2 and Fig. 3 show the results. There are big differences between before and after replacing. Most of the correlation patterns disappear in Fig. 2 (b), (d), (f) and (h) after replacing. This indicates that top and bottom 5% extreme precipitation plays important roles to produce the correlation patterns in our study.

Corrections:

(Line 126-130) To further prove that extreme rainfall is significantly influenced by these important regions, top and bottom 5% extreme PA is replaced by the random middle magnitude PA in data. Then we employ the same analysis of the new time series. Fig. 2(b), (d), (f) and (h) shows the results after replacing. Comparing with Fig. 2(a) before replacing, some important nodes disappear in Fig. 2(b) after replacing. Similar results also can be found for other seasons. It implies that top and bottom 5% extreme precipitation plays important roles to contribute to the teleconnection patterns.

*3. The manuscript needs a substantial improvement on the language. The manuscript is very hard to follow and understand, many sentences are ambiguous. Figures and legends are unclear and misleading. The description of the study approach is incomplete and an in-depth discussion from both statistical and physical perspectives is missing in the manuscript.*

Response: We thank the referee for the helpful comment. We did our best to improve the revised manuscript. The figures, methods and discussions were revised based on the referees' comments.

*Major and specific problems that must be addressed before reconsideration are attached below:*

*Major problems must be addressed:*

*1. Introduction: The 2$^{nd}$ and 3$^{rd}$ paragraphs in the introduction are literature review on rainfall in SWC and applications of complex network methods respectively, however, both of them just list several related studies without a logical construction. It is quite difficult for readers to follow up, and it does not help leading to the specific research questions of this study. Furthermore, at the end of this paragraph (i.e., Lines 29-31), authors claim that "most of studies discussed the rainfall of the SWC only for single season", therefore this study would like to explore the relation in different seasons. This is not true. There are many studies done by both Chinese scholars and overseas scholars on different seasons, even if different studies might have focus on one or two seasons. So I recommend the authors to remove this claim. I strongly suggest the authors to revise the literature review, most importantly to include appropriate literature for the scientific part (research gaps etc.) and for the method they mainly adopt (complex network).*

Response: We thank the referee for this kind and valuable comment. We improved the introduction in the revised manuscript by a logical way. In the 1st paragraph, we introduced the importance of NWSA in research fields and explained why we need to study it. In the 2nd paragraph we introduced some research progress on precipitation in NWSA. Then we extended literature reviews on complex networks in the 3rd paragraph. We also removed the disputable statements in the revised manuscript.

Corrections:

(Line 12-19) In recent decades, natural hazards (such as droughts and floods) have occurred frequently in Northwestern South Asia (NWSA) due to climate change, causing a large number of casualties and property losses (Ha et al., 2019; Gao et al., 2017; Wei et al., 2018). In the summer of 2006 and 2011, NWSA suffered from record-breaking droughts events (Zhang et al., 2017). On the other hand, the portion of annual precipitation contributed by extremely heavy precipitation has been found an increasing trend from 1961–2010 in NWSA (Ma et al., 2013). Due to an increasing population and the high risk of natural hazards, NWSA has attracted lots of attention in meteorological research fields. According to CMIP5 multi-model projections, they found that severe and extreme droughts in NWSA increase dramatically in the future, and extremely wet events will also increase (Wang et al., 2014).

(Line 20-32) Droughts and floods can be attributed to precipitation anomalies. A better understanding of precipitation can further improve the underlying mechanisms of droughts and floods in NWSA. Annual precipitation over NWSA did not show a significant decreasing or increasing trend (Qin et al., 2010; Zhang et al., 2017). In fact, the trend of precipitation in NWSA has been found to strongly depend on seasons and locations (Shi et al., 2015; Huang et al., 2014; Ma et al., 2013). Precipitation in NWSA is very difficult to forecast, since it can be influenced by both the East Asian monsoon and Indian monsoon that carry moist air from the Indian Ocean and

Pacific Ocean to this region (Qian et al., 2002; Renhe et al., 2001). Wang et al. (2015) investigated the inter-annual variation of autumn precipitation in NWSA and found the teleconnection between tropical Northwest Pacific sea surface temperature and autumn precipitation in NWSA. Furthermore, autumn precipitation in NWSA experienced a notable wet-to-dry shift in 1994 that could be influenced by the tropical warm pool (Wang et al., 2018). During the drought in NWSA in 2006, the western Pacific subtropical high was abnormally high, which can inhibit the water vapor transport to NWSA resulting in extremely low precipitation (Li et al., 2011). Feng et al. (2014) studied the drought events in NWSA from 1951 to 2010 and found that most of these events were related to the tropical Pacific and North Atlantic sea temperature anomalies.

2. *Line 60: As this study discusses the relation between rainfall and SSTA for four seasons independently, why the authors still need removal of the seasonal cycle of rainfall and SSTA data? Also, in fact, I am not clear how the authors did the removal. Please clarify.*

Response: We thank the referee for the comments. Because we still saw a seasonal trend in the 3-months' time series for non-removal of the seasonal cycle as showed in below Fig. 10, especially for the SSTA. After we detrended it, this seasonal trend can be removed. We explained the details how we did the removal in the revised manuscript.

Corrections:

(Line 60-65) First, we remove the seasonal cycle to obtain the time series of the SSTA as (Fan et al., 2017; Meng et al., 2017),

$$Y^y(t) = \frac{\widetilde{Y}^y(t) - \text{mean}\left(\widetilde{Y}(t)\right)}{\text{std}\left(\widetilde{Y}(t)\right)}, \quad (1)$$

where $\widetilde{Y}^y(t)$ is the time series of the daily SST; y stands year and t stands date within a year. "mean" and "std" denote the mean and standard deviation of the SST for all the years on a date t. We use the same way to obtain precipitation anomalies (PA).

3. *Line 75-80: Why should we separate the positive and negative degree? Since the authors only mention that they are different characteristics and display the corresponding regions for positive and negative degrees, but the explanation for the underlying mechanisms of these two degrees is lacking. For instance, in Line 100-110, the authors state that most of the clusters locating in the tropics are reasonable because of the important role of Hayley circulation in moisture transportation. But how does Hayley circulation involve in both positive and negative degrees? The authors need to provide clear explanation otherwise it is difficult for readers to follow. And how do the positive and negative degrees contribute to the rainfall in SWC?*

Response: We thank the referee for the helpful comment. Negative degrees can reflect the information about anti-correlation to be different with that of the positive degrees. Thus we considered them separately. Hayley circulation is a general circulation to connect the equator and non-equator. To be specific, we improved our explanation that the mechanisms are related to the East Asian monsoon and Indian monsoon which can carry moist air from the Indian Ocean and Pacific Ocean to NWSA resulting in extreme precipitation. Some of the clusters in the topics can directly influence on precipitation in NWSA. Other clusters are indirectly connected to NWSA. So the anti-correlation of the SSTA itself between two SSTA areas can result in positive and

negative degrees in the two different places. We improved the explanation in the revised manuscript.

Corrections:

(Line 126-130) Due to the special geographical location of NWSA, both the East Asian monsoon and Indian monsoon can carry moist air from the Indian Ocean and Pacific Ocean to NWSA resulting in extreme precipitation. Thus we suggest that these important out-degree areas and NWSA are connected by the East Asian monsoon and the Indian monsoon. Different features of the monsoon account for the changes of the degree patterns in different seasons.

(Line 135-137) In fact, there are correlations for the SSTA itself between two regions i.e. the SSTA in the east Equatorial Pacific is negative correlated with the SSTA in the west Equatorial Pacific. Such anti-correlations can lead to the positive and negative correlation patterns between the PA in NWSA and SSAT in the two different places.

4. *Lines 82-88: I suggest the authors to clearly state methodology in details in Section 2.2 instead of in the result section. In addition, the authors should pay attention to address the following questions in their revised manuscript: (1) how to obtain the shuffled data? (2) this sentence, "no correlation between the shuffled time series" (Lines 85-86), is quite confusing. Does it mean "no significant autocorrelation for one time series" or "no significant correlation between two different shuffled time series"? (3) it is also unclear why this threshold (i.e., 0.11) is appropriate. At least some sensitive test should be provided to see the effect of choosing 0.11 or close values. It seems that values around 0.1 are all reasonable guess based on Figure 1. Also, I suggest to reorganize the result section into several subsections in line with revised methodology part, in order to have a clearer structure.*

Response: We thank the referee for these valuable comments. We improved the method to obtain shuffled data as Fan et al., (2017) did. We randomly shuffled the order of years, keeping the variations within each year and then calculated the cross-correlation between the shuffled time series. In Fig. 1, we showed the PDF of correlations for the shuffle data comparing with real data. In this shuffling approach, the distribution of values and the autocorrelations in each year have been kept in each shuffled record, and the physical dependencies between nodes tend to be destroyed. If the correlations are significantly higher than the significant threshold, we regarded it as a true link; otherwise, it is suspected to be a spurious link. We obtained the threshold $\Delta = 0.1$ by using the 95% confidence significance test combined with a multiple testing correction (Benjamini-Hochberg). We improved and reorganized the result into three subsections in the revised manuscript: 1. Significance tests, 2. Links and degrees and 3. Important areas.

Corrections:

(Line 99-107) In order to verify the significance of the correlation, we compare the PDFs between the real data (red) and shuffled data (blue) in Fig. 1. We randomly shuffle the order of years for each node, keeping the variations within each year to get shuffle data (Fan et al., 2017). Then we calculate the cross-correlation for shuffled data as same as real data. In this shuffling process, the autocorrelations and common seasonality in each year have been kept in each shuffled time series, while the physical dependencies between the SSTA and PA nodes are destroyed. The PDF of real data in Fig. 1 shows a much slower decay than that of shuffle data with the increased absolute correlation |C| both in the positive and negative parts. If the correlations are significantly higher than the significant threshold $\Delta$, we regard it as a true link; otherwise, it is suspected to be a spurious link as Eq. 3. We obtain the threshold $\Delta = 0.1$ by using

the 95% confidence significance test combining with a multiple testing correction (Benjamini-Hochberg) (Thissen et al., 2002).

5. *Line 88-90: the authors try to verify the significance of the correlation through comparing the PDFs for the real data and shuffled data. However, the difference is not that distinct, with the maximum correlation of the real data only reaching 0.2 compared to 0.1 for the shuffled data. Besides, how is the data being shuffled? Just randomize the whole original time series? Or as done in Fan et al. (2017), the time series is shuffled only in year level and the time ordering within a year is unchanged.*

Response: We thank the referee. Please see above response. The correlation $C = 0.1$ is already larger than most of the correlations (95%) for shuffle data. Thus the maximum correlation $C = 0.2$ is very significant.

6. *Figure 2: The maps are too small for the reader to interpret. The same problem is with Figure 3, Figure 6 - 8 (a, c) and Figure 9 (a). In addition, the caption of Figure 2 is not consistent with color bar.*

Response: Thank the referee. We corrected the errors in the revised manuscript.

7. *L99 – 100: Why is the regional size in spring greater than other seasons? I suggest some explanation should be provided. In fact there are many places that the authors only present the observation from figures without extended discussion or explanation. It is very important to provide insights rather than just purely stating patterns that can be seen from the figures.*

Response: We thank the referee for the helpful comment. We suggested that greater regional size in spring could be caused by the El Niño. Most of the influence areas were found in the Pacific. Seeing Fig. 7 (b), (d), the time delay between the west (east) Pacific and SWC is nearly 3 months, since El Niño events probably influence NWSA in MAM according to the monsoon as a mediator three months after the SSTA in the east Pacific reaches to the peak on the December of last year. We extended the discussion in the revised manuscript.

Corrections:

(Line 165-171) For MAM, there are most links between the SSTA and PA in NWSA. The largest cluster size is also larger than other seasons. Figs. 7(a) shows that the cluster $C_1$ in the western Pacific and $C_2$ in the Equatorial Atlantic are positively correlated with the nodes $I_{11}$ in MAM. For the negative correlation, we find the clusters $C_1$ and $C_2$ in the east Pacific and Indian Ocean (see Figs. 7(c)). Thus most of the correlation patterns come from the Pacific. Moreover, the time delay of the strongest positive and negative link for $C_1$ are within 100 days shorter than that of DJF, but the strength of links is stronger seeing Figs. 7(b) and (d). Thus we suggest that El Niño events probably influence the PA of $I_{11}$ in MAM three months after the SSTA in the east Pacific reaches to the peak on the December of last year during El Niño. Next winter PA of $I_{11}$ is influenced by the El Nino with nearly one year delay.

8. *Figure 4 and 5: The legend of color bar is missing. The descriptions of these figures are quite confusing (Lines 110 - 119). For instance, $C_3$ is the grid with the largest value in the Figure 4 (c), then why it is chosen there? As in the manuscript, $C_1$ to $C_4$ are selected based on the largest in-degree value ($C_1$ is consistent with Figure 4, but I am not sure about $C_4$). The same problem is*

*also found in Figure 5 (d) – inconsistent with the manuscript. Authors should clarify their statement carefully.*

Response: We thank the referee for the helpful comment. We corrected these and redefined the grid with the largest in-degree as $I_x$ in Figs. 4 and 5 for each season in the revised manuscript.

9. *Similar to Comment # 5: L113 – 115: What are the relationships between the identified nodes spatial patterns and the inhomogeneous spatial distribution of rainfall in SWC? The authors could elaborate more about the spatial distribution of rainfall in SWC.*

Response: Thank the referee very much. We tried to find the relationships between the identified nodes spatial patterns and the spatial distribution of rainfall in NWSA. Unfortunately, there no exits a sensible relation. So we removed this sentence in the revised manuscript. Instead, we suggested that the significant in-degree nodes are dependent on their geographical locations.

Corrections:

(Line 140-144) A lot of nodes within NWSA show a small in-degree value. Only several nodes have strong correlations with the SSTA as shown in Figs. 4 and 5. The distribution of in-degrees is localized and change with seasons. The important nodes with the large in-degree values are almost found in the left- and right- bottom corners of Figs. 4 and 5 which are close to the Indian and west Pacific Ocean respectively. Thus these nodes of NWSA are easier to be influenced by the different monsoons than other nodes in NWSA. Also seasonality of the in-degree could be attributed to the monsoons.

10. *L115 – 118: What are the possible mechanisms that induces the changes of the spatial distributions of identified nodes with seasons? E.g., the joint-effects of terrain and important SSTA nodes.*

Response: We thank the referee for the comment. The possible mechanisms are related to the East Asian monsoon and Indian monsoon. Due to the special geographical location of NWSA, both them can carry moist air from the Indian Ocean and Pacific Ocean to NWSA resulting in extreme precipitation. Different features of the monsoons account for the changes of the degree patterns in different seasons. Also please see above corrections.

Corrections:

(Line 121-125) Due to the special geographical location of NWSA, both the East Asian monsoon and Indian monsoon can carry moist air from the Indian Ocean and Pacific Ocean to NWAS resulting in extreme precipitation. Thus we suggest that these important out-degree areas and SWC are connected by the East Asian monsoon and Indian monsoon (Zhao et al., 2009; Gong et al., 2018; Zhang et al., 2017; Feng et al., 2014). Different features of the monsoons account for the changes of the degree patterns in different seasons.

11. *L118 – 119: The sentence may be inappropriate, please rewrite it. Since one node of SSTA may positively and negatively correlate with different nodes in SWC.*

Response: We thank the referee very much. We rewrote it as below.

Corrections:

(Line 144-146) Furthermore, the largest positive and negative weighted in-degree nodes are same for MAM (see Figs. 4(b) and 5(b)). This indicates that this node is positively (negatively) correlated with the most SSTA areas.

12. *Lines 121-125: A significant test for correlations much be done, as the absolute values of correlation in Figure 6 (b, d) is only around 0.1. With all these very weak correlation values, I cannot be convinced by the statement such as "a high daily SSTA in East Equatorial Pacific is probably observed ... in SWC". The same problem is also found in the discussion for different nodes (Lines 135-144). And the color bars of Figure 6, 7, 8 (a, c) and Figure 9 (a) are incomplete.*

Response: We thank the referee for the comment. We already did the significance tests as addressed in comment #4. We also replaced to show Figures 6-9 (b, d) by using the strongest links that are clear. The color bars were corrected.

13. *Figure 6 a&c: I think there are better ways to select critical SSTA regions, rather than just simply comparing Figure 2 and Figure 3. I think the authors could utilize more advanced method (e.g. in [Kawale, 2013; Lu et al., 2016])*

Response: We thank the referee for the helpful comment. We read and use the mentioned method [Kawale, 2013; Lu et al., 2016] to identity clusters. In section 3.3 of the revised manuscript, we identified the two largest cluster for the SSTA areas connected to the important node in NWSA.

Corrections:

(Line 149-153) We first select the nodes in NWSA with the largest weighted in-degree for each season as showed in Figs. 4 and 5. The largest cluster $C_1$ is identified by the largest successive area where all the inside SSTA nodes are connected to that important node in NWSA (Kawale, 2013; Lu et al., 2016). We can obtain the second largest cluster $C_2$ in a similar way. Figs. 6(a) shows the cluster $C_1$ (blue) and $C_2$ (green) which are connected to the nodes of $I_{11}$ (as shown in Fig. 4(a)) for DJF.

14. *L141 – 144: Can the authors provide the links between nodes $C_2$, $C_3$ and the SSTA nodes for both MAM and JJA. Because both $C_2$ and $C_3$ are important nodes in spring and summer based on figures 4 and 5. The authors should explain more if the SSTA nodes affecting $C_2$ are different with the nodes affecting $C_3$ in spring or summer.*

Response: We thank the referee for the valuable comment. Note that we renamed the nodes $C_2$ as $I_{11}$ and $C_3$ as $I_{31}$ in the revised manuscript. In below Fig. 11, we provided the SSTA nodes which are connected to $I_{11}$ and $I_{31}$ for MAM and JJA respectively. Interestingly, we found that the connected SSTA areas of $I_{11}$ and $I_{31}$ are very different in MAM for the positive correlation. (see Fig. 11 (a) and Fig. (e)). $I_{11}$ is linked to the west Pacific, but $I_{31}$ is linked to the east Pacific and the Indian ocean. The big differences are also observed for the negative correlation (in Fig. 11(c), (g)) and in JJA (Fig. 11(b), (d), (f), (h)), although the distance between $I_{11}$ and $I_{31}$ is not long. We have not yet fully understood the mechanism. But we thought that the mechanism should be related to the special climate in NWSA. There are high mountains between the two places leading to the non-similarity. And the places could be controlled by the different monsoon systems.

15. *Conclusion part: This section is only a brief summary of the study. The authors are expected to provide an in-depth discussion from the physical perspective, like potential mechanism, instead of just listing some related results from previous studies. I do not see any contribution from this study from reading the conclusion part.*

Response: We thank the referee for the valuable comment. We improved our conclusion part as follows.

Corrections:

(Line 178-185) In summary, we employ a multi-variable complex network method to study the teleconnection between the SSTA and PA in NWSA. We show the most teleconnection links in spring, followed by winter. There are less links in summer and autumn. The El Niño could cause the stronger teleconnections in spring. According to the weighted out-degree of the network, we show that the positive and negative correlation patterns over the world are mainly contributed by the extreme PA in NWSA. Moreover, We find that most of the out-degree patterns emerge in the Equatorial Indian, Pacific and Atlantic Oceans. The mechanisms are related to the East Asian and Indian monsoons for the Equatorial Indian and Pacific Oceans. For the Atlantic, long-range planetary waves account for the teleconnections. Due to seasonality of the monsoons, we also find that the out-degree patterns significantly change with season.

(Line 186-192) According to the weighted in-degree in NWSA, we find that the teleconnections are dominated by the several specific nodes within NWSA. These nodes are closer to the oceans than other nodes. Thus we focus on the important nodes in NWSA to obtain the two largest clusters which are both connected to one of the important nodes in NWSA. The time delay between the cluster and the node in NWSA is given by the strongest link. We find that the largest cluster in the east Pacific for the node $I_{11}$ in NWSA could be related to the El Niño with the time delays −230 and −79 days for DJF and MAM respectively. These results could be useful to improve the prediction of rainfall in NWSA. In future work, we will focus on the prediction based on these observed teleconnections.

16. *When I read the abstract (Lines 9-10), I got interested in the study because the authors claimed that "the time-lag of the teleconnection links ... prediction of rainfall in SWC". After I read this manuscript, I do not see how this study could achieve this, the authors should provide related analysis or discussion to support how this study can improve the rainfall prediction.*

Response: We thank the referee very much. Rainfall prediction in NWSA is a very difficult issue. It will be very important even for a small progress. In the present study, we cannot claim that we improved the prediction. But the time delays between important SSTA areas and NWSA could be helpful to improve the prediction. Previous studies for NWSA didn't show these. In further study, we will focus on the prediction based on these results.

Corrections:

Abstract: Droughts and floods have frequently occurred in Northwestern South Asia (NWSA) in this century. The mechanism of precipitation in NWSA is quite complicated, since the East Asian monsoon, Indian monsoon and et al. potentially influence the rainfall in this region. Prediction of precipitation in NWSA has become a difficult and critical topic in climatology study. Thus we here develop a novel multi-variable network method to delineate the relations between the global sea surface temperature anomalies (SSTA) and the precipitation anomalies (PA) in NWSA. Our results show the important out-degree patterns in the Pacific, Atlantic and Indian Ocean, which significantly influence the PA in NWSA. Particularly most of the patterns are caused by extreme

precipitation and change with the season. Furthermore, the in-degree patterns indicate that the teleconnections are dominated by several important nodes within NWSA. According to study these nodes in NWSA, we find that the SSTA areas influence the nodes in NWSA with some specific time delays (more than 50 days) that could be helpful to improve long-term prediction of precipitation in NWSA.

*Some minor issues:*
*1.  L128: I suggest removing ", which has been closed to the limit of the time lag" unless the authors can evaluate the significance of it.*

Response: We thank the referee for the helpful comment. We did it and extended the range of the time lag (see Fig 6-9) in the revised manuscript.

[Figure]

**Figure 1.** (Color online) PDFs of correlations $C_{ij}$ for real data and shuffle data. Black vertical lines represent the location of the threshold $|\Delta| = 0.1$.

[Figure]

**Figure 2.** (Color online) Distributions of the positive weighted out-degree for (a) DJF, (c) MAM, (e) JJA and (h) SON. (b), (d), (f) and (h) Same as (a), (c), (e) and (h) but for replacing top and bottom 5% extreme precipitation with middle magnitude precipitation in data. White areas represent zero in maps. Purple rectangle area covers the region of NWSA.

[Figure]

**Figure 3.** Same as Fig. 2 but for the negative weighted out-degree.

[Figure]

**Figure 4.** (Color online) Distributions of the positive weighted in-degrees for (a) DJF, (b) MAM, (c) JJA and (d) SON in SWC. The location of NWSA in the world is shown as the purple rectangle area in Fig. 2. $I_{11}$, $I_{31}$ and $I_{36}$ are the important nodes in NWSA with the largest positive weighted in-degree in a season.

[Figure]

**Figure 5.** (Color online) Same as FIG. 4 but for the negative weighted out-degree. $I_{15}$, $I_{11}$, $I_{22}$ and $I_{12}$ are the important nodes in NWSA with the largest negative weighted in-degree in a season.

[Figure]

**Figure 6.** (Color online) Locations of the largest cluster $C_1$ (blue) and the second largest cluster $C_2$ (green) that are (a) positively ((c) negatively) correlated with the node $.I_{11}$, ($I_{15}$) in NWSA for DJF. The blue and green arrows represent the strongest links from $C_1$ and $C_2$ to that node in NWSA respectively. (b), (d) The correlation $\hat{C}$ as a function of the time lag $\tau$ corresponding to the strongest links in map (left) respectively. Dashed black line shows the absolute maximum of the correlation $\hat{C}$.

[Figure]

**Figure 7.** (Color online) Locations of the largest cluster $C_1$ (blue) and the second largest cluster $C_2$ (green) that are (a) positively ((c) negatively) correlated with the node $I_{11}$ ($I_{11}$) in NWSA for MAM. Everything else is the same as Fig. 6.

[Figure]

**Figure 8.** (Color online) Locations of the largest cluster $C_1$ (blue) and the second largest cluster $C_2$ (green) that are (a) positively ((c) negatively) correlated with the node $I_{31}$ ($I_{22}$) in NWSA for JJA. Everything else is the same as Fig. 6.

[Figure]

**Figure 9.** (Color online) Locations of the largest cluster $C_1$ (blue) and the second largest cluster $C_2$ (green) that are (a) positively ((c) negatively) correlated with the node $I_{36}$ ($I_{12}$) in NWSA for SON. Everything else is the same as Fig. 6.

[Figure]

**Figure 10.** The non-detrended time series of (a) the SSTA and (b) PA in 3 months.

[Figure]

**Figure 11.** (Color online) The SSTA areas which are connected to the nodes $I_{11}$ (a-d) and $I_{31}$ (e-f) in MAM and JJA respectively. Color bars represent the correlations of the links.